# Role of leukotriene B$_4$ (LTB$_4$)-LTB$_4$ receptor 1 signaling in post-incisional nociceptive sensitization and local inflammation in mice

**Miho Asahara[1], Nobuko Ito[1]\*, Yoko Hoshino[1], Takaharu Sasaki[2], Takehiko Yokomizo[2], Motonao Nakamura[3], Takao Shimizu[4,5], Yoshitsugu Yamada[6]**

**1** Department of Anesthesiology, Faculty of Medicine, The University of Tokyo, Tokyo, Japan, **2** Department of Biochemistry, Juntendo University School of Medicine, Tokyo, Japan, **3** Department of Life Science, Faculty of Science, Okayama University of Science, Okayama, Japan, **4** Department of Lipid Signaling, National Center for Global Health and Medicine, Tokyo, Japan, **5** Department of Lipidomics, Faculty of Medicine, The University of Tokyo, Tokyo, Japan, **6** International University of Health and Welfare, Mita Hospital, Tokyo, Japan

\* nobuko.tky@gmail.com

**Data Availability Statement:** All relevant data are within the paper.

## Abstract

Leukotriene B$_4$ (LTB$_4$) is a potent lipid mediator involved in the recruitment and activation of neutrophils, which is an important feature of tissue injury and inflammation. The biological effects of LTB$_4$ are primarily mediated through the high-affinity LTB$_4$ receptor, BLT1. Postoperative incisional pain is characterized by persistent acute pain at the site of tissue injury and is associated with local inflammation. Here, we compared the role of LTB$_4$-BLT1 signaling in postoperative incisional pain between BLT1-knockout (BLT1KO) and wild-type (BLT1WT) mice. A planter incision model was developed, and mechanical pain hypersensitivity was determined using the von Frey test before and after incision. Local infiltration of neutrophils and inflammatory monocytes was quantified by flow cytometry. Inflammatory cytokine levels in the incised tissue were also determined. Mechanical pain hypersensitivity was significantly reduced in BLT1KO mice compared to BLT1WT mice at 2, 3, and 4 days after incision. LTB$_4$ levels in the tissue at the incision site peaked 3 hours after the incision. Infiltrated neutrophils peaked 1 day after the incision in both BLT1KO and BLT1WT mice. The accumulation of inflammatory monocytes increased 1–3 days after the incision and was significantly more reduced in BLT1KO mice than in BLT1WT mice. In BLT1KO mice, Interleukin-1β and Tumor Necrosis Factor-α levels 1 day after the incision were significantly lower than those of BLT1WT mice. Our data suggest that LTB$_4$ is produced and activates its receptor BLT1 in the very early phase of tissue injury, and that LTB$_4$-BLT1 signaling exacerbates pain responses by promoting local infiltration of inflammatory monocytes and cytokine production. Thus, LTB$_4$-BLT1 signaling is a potential target for therapeutic intervention of acute and persistent pain induced by tissue injury.

**Funding:** This study was supported by Grants in Aid for Scientific Research, Japan Society for Promotion of Science (JSPS) in the form of grants awarded to NI (20K09190) and TY (18H02627 and 21H04798). This study was also supported by AMED in the form of a grant awarded to TY (22wm0425008s0201). The funders had no role in this study design, data collection and analysis, decision to publish, or preparation of the manuscript.

**Competing interests:** The authors have declared that no competing interests exist.

## Introduction

Leukotriene B$_4$ (LTB$_4$), a metabolite of arachidonic acid, is a potent proinflammatory lipid mediator that activates and recruits leukocytes to inflamed regions [1]. The potent biological effects of LTB$_4$ are mediated primarily through LTB$_4$ receptor 1 (BLT1), which is a high-affinity G protein-coupled receptor for LTB$_4$ [1, 2]. BLT1 is expressed in several types of leukocytes, which include neutrophils, macrophages and their precursors [2], monocytes [3], differentiated T cells [4, 5] and dendritic cells [6]. We and others have established BLT1-knockout (BLT1KO) mice [7, 8], and previous studies using BLT1KO mice showed that LTB$_4$-BLT1 signaling is involved in a variety of inflammatory and immune responses, including allergic airway inflammation [4, 7, 9, 10], multiple sclerosis [11], atherosclerosis [3, 12, 13], inflammatory arthritis [14, 15], tumor promotion [16], and psoriasis [17]. Specifically, regarding the inflammatory model, numerous studies have shown that the blockade of LTB$_4$-BLT1 signaling significantly inhibits neutrophil recruitment and inflammatory conditions [17–19].

In the 1980s, Levine et al. reported that intradermal injection of LTB$_4$ evokes thermal and mechanical allodynia [20]. Subsequently, several studies demonstrated the involvement of LTB$_4$-BLT1 signaling in the progression of inflammatory pain. The expression of BLT1 in murine dorsal root ganglion and the spinal cord has been confirmed [21–23]; moreover, BLT1 mRNA levels were increased in rat spinal cord neurons in a spared nerve injury model of neuropathic pain [22]. Our previous study showed that BLT1KO mice exhibited reduced local inflammation and marked attenuation of acute nociception induced by intraplantar formalin injection, which suggested that LTB$_4$-BLT1 signaling is involved in the mechanism of inflammatory pain [19].

Tissue injury causes the release of various chemical mediators from injured tissues and immune cells, which initiates the inflammatory response and sensitizes peripheral nociceptors [24–28]. In the early phase of inflammation, neutrophils are the most dominant immune cells, which migrate through the vascular endothelium, accumulate in the injured tissue, and stimulate the release of various nociceptive mediators, such as cytokines, chemokines, and other substances [29, 30]. This initial response is followed by the recruitment of monocytes to inflammatory sites and the activation and proliferation of tissue-resident macrophages, which occur several days after tissue injury [31]. Furthermore, inflammatory monocytes and macrophages release nociceptive mediators and expand the neutrophil response. Thus, tissue injury activates various types of immune cells in chronological order, and nociceptive mediators released from these immune cells support and prolong pain responses, as shown in rodent incisional pain models [32–38].

Therefore, in this study, we focused on the pain associated with tissue injury and investigated the role of LTB$_4$-BLT1 signaling. As a popular tissue injury pain model, we used the hind paw incisional pain model, which is defined as a preclinical animal model of postoperative pain. We also determined the immune cells, and the cytokine production processes that were affected by BLT1 deficiency at the site of tissue injury during the course of incisional pain.

## Materials and methods

### Animals

BLT1KO mice were generated and bred on a C57BL/6 background as described previously [39]. All mice were maintained on a 12-hour light/dark cycle with ad libitum access to water and food. This study was carried out in accordance with the University of Tokyo's guidelines for the care and use of laboratory animals. The protocol was approved by the Animal Experimental Committee of the University of Tokyo (P17-025). All surgery was performed under anesthesia, and all efforts were made to minimize suffering.

## Plantar incision

The hind paw plantar incision mouse model was created as previously described with minor modifications [28]. Briefly, mice were anesthetized by inhalation of isoflurane via a nose cone. After sterile preparation, a 5-mm longitudinal incision was made through the skin and fascia of the plantar aspect of the right hind paw with a No. 11 scalpel. The underlying plantaris muscle was elevated and incised longitudinally, allowing the muscle origin and insertion to remain intact. After hemostasis with gentle pressure, the skin was opposed with two sutures using 8–0 nylon threads, and the wound was covered with antibiotic ointment. The mice were then allowed to recover in their home cages.

## Behavioral testing

Mice were acclimated to testing cages containing a stainless steel mesh for 1hr per day at least 2 days before testing. Mechanical pain hypersensitivity was estimated using von Frey filaments according to the up-down method, as described previously [40]. Briefly, mice were placed individually on wire mesh platforms in clear cylindrical plastic enclosures and allowed to acclimate to the environment for 30 minutes prior to the von Frey test. Von Frey filaments were applied to the middle of the plantar surface of the hind paw. A series of six von Frey filaments (0.7-, 1.6-, 4-, 6-, 20-, and 40-mN forces) were used. Testing was initiated with 20 mN forces. Whenever a positive response occurred, the next weaker von Frey filament was applied. Whenever a negative response occurred, next stronger one was applied. The test was continued, until the response of six stimuli after the first change in response had been obtained or the test reached either end of the spectrum of the von Frey set. Assessments were performed before surgery (baseline), 2 hours after surgery, and on postoperative days 1, 2, 3, 4, 5, and 7.

Withdrawal responses following stimulation were measured, and a tactile stimulus that produced a 50% likelihood of withdrawal was determined. Mechanical Threshold = $(10^{[Xf+k\delta]})/10,000$, where $Xf$ = value (in log units) of the final von Frey filament used; $k$ = tabular values for the pattern of positive/negative responses; and $\delta$ = mean difference (in log units) between stimuli (here, 0.38).

Spontaneous pain related behaviors following LTB₄ administration were assessed using guarding pain score (GPS) of the hind paws according to a previously described method [41]. Immediately after injection of LTB₄ or vehicle, mice were placed individually on a stainless-steel mesh floor (openings, 8 X 8 mm) under a clear plastic cage. GPS was calculated on the basis of weight bearing. Both paws of each animal were closely observed during a 1-min period repeated every 5 min for 30 min. Depending on the position in which the paw was found during the scoring period, a score of 0, 1 or 2 was given. 0: both foot full touch 1: partial 2: complete off. The sum of six scores (0 to 12) was used to assess the spontaneous pain following LTB₄ injection to the hind paw. The person performing behavioral tests was blinded to the treatment and genotypes of mice.

## LTB₄ administration in the hind paw

LTB₄ (20110, Cayman Chemical, MI, USA) injection for assessments of mechanical responses were performed under light isoflurane anesthesia using 4% (1 min) for induction and 1.5% (1 min) for maintenance during injection. LTB₄ at dose of 10ng, 1ng in 5μl PBS or vehicle was injected into plantar surface of the hind paw subcutaneously using a Hamilton syringe attached to a 30-G needle in a total volume of 5 μl. For spontaneous pain behaviors, injections were performed using gentle restraint without anesthesia. The needle tip was inserted between digits and directed proximally, where drug or vehicle was injected into the middle of the hind paw. To minimize needle trauma, needle tip did not invade the area that mechanical stimuli were applied. The person performing the behavioral experiments was blinded to the drug and dose.

## Quantification of LTB$_4$ concentration in the paw

Plantar tissue samples, including skin and underlying muscle were collected at different time points after plantar incision under deep anesthesia with pentobarbital. Each sample was diluted with a 200-μL solution of methanol: formic acid (100:0.2) by centrifugation (at 100 g for 1 min). The column was washed with 0.1% formic acid solution, 15% methanol containing 0.1% formic acid, and petroleum ether containing 0.1% formic acid. All samples were analyzed using liquid chromatography-tandem mass spectrometry (LC-MS/MS) as described previously [42].

## Flow cytometric analysis

Evaluation of leukocyte infiltration was performed as previously described with minor modifications [43, 44]. Plantar tissue samples, including skin and underlying muscle were first dissected, as described in the previous section. The samples were then minced and digested in a mixture of 1 mg/mL collagenase A (Roche, Mannheim, Germany) plus 2.4 U/ml Dispase II (Roche, Mannheim, Germany) in Hanks' Balanced Salt solution (Thermo Fisher Scientific, Waltham, MA, USA) at 37˚C for 2 hours. Following digestion, the cell suspension was washed in Flow Cytometry staining (FACS) buffer (eBioscience, San Diego, CA, USA), filtered through a 70-μm mesh, and resuspended in FACS buffer. The filtrate was suspended in FACS buffer to produce a single-cell suspension and blocked with Fc Receptor blocking reagent (1:10, Miltenyi Biotec, Tokyo, Japan) on ice for 10 min. For surface receptor labeling, cells were incubated with mixtures of the following antibodies in the dark for 30 min on ice: anti-CD11b-PE (1:800, monoclonal, rat, 130-091-240, Miltenyi Biotec, Tokyo, Japan), anti-Ly6G-APC (1:200, monoclonal, rat, 127614, Biolegend, San Diego, CA,USA), and anti-Ly6C--FITC (1:200, monoclonal, rat, 128006, Biolegend, San Diego, CA, USA). Dead cells were gated out according to propidium iodide (Biolegend, San Diego, CA, USA) staining, and isotype controls were used to titrate each antibody to minimize background staining. Flow cytometry was conducted on an Accuri™ C6 flowcytometer (Becton Dickinson and Company, NJ, USA). Compensation was performed at the beginning of each experiment. For all samples, 50,000 cells were analyzed to generate scatter plots. Cell suspensions were gated on forward-scatter (FSC-A) and side-scatter areas (SSC-A) to exclude debris. Cells that were double-positive for CD11b and Ly6G were recognized as neutrophils. Non-neutrophil myeloid cells (CD11b$^+$Ly6G$^-$) were then gated according to Ly6C expression as Ly6C$^{high}$ monocytes (inflammatory monocytes) or Ly6C$^{low}$ monocytes (resident monocytes). All data were analyzed using FlowJo software (Becton Dickinson and Company, NJ, USA).

## Cytokine analysis

Tissue cytokine levels were measured using a method similar to that described previously [32]. Briefly, plantar tissue samples, including skin and underlying muscle were dissected as described in the previous section and placed immediately in ice-cold 0.9% sodium chloride that contained a cocktail of protease inhibitors (Complete; Roche, Mannheim, Germany). Samples were then homogenized and centrifuged at 12,000 g at 4˚C. The supernatant fractions were stored at −80˚C until use.

For cytokine analysis, BD$^{TM}$ Cytometric Bead Array (CBA) Mouse Cytokine Kits (BD Biosciences, San Jose, CA, USA) were used for the simultaneous detection of mouse interleukin-6 (IL-6), interleukin-1β (IL-1β), and tumor necrosis factor-α (TNF-α) in a single sample. In accordance with manufacturer recommendations, three capture beads with distinct fluorescence intensities, coated with cytokine-specific capture antibodies, were mixed in equal volumes: 50 μL of each sample and 50 μl of Phycoerythrin-conjugated detection antibodies were

added to 50 μL of mixed-bead populations. These capture beads were incubated with recombinant standards or tissue samples for 1 hour at room temperature in the dark to form sandwich complexes.

The samples were washed with wash buffer, and the bead pellet was resuspended in 300 μL wash buffer after discarding the supernatant. Samples were measured on a Flow Cytometer (Accuri™ C6) and analyzed using FCAP Array™ Software. Individual cytokine concentrations were indicated by their fluorescent intensities. Cytokine standards were diluted to facilitate the construction of calibration curves, which were required to determine protein concentrations of the test samples.

## Statistical analysis

Statistical analysis was performed using Prism 6.0 (Graph Pad Software, La Jolla, CA, USA) and R statistical software version 4.2.1 (R Project for Statistical Computing). Two-way analyses of variance (ANOVA) and Bonferroni post hoc tests were used to analyze the behavioral tests, infiltrated neutrophils immune cells, and local cytokines. Analysis of local LTB$_4$ concentration and Guarding pain score were performed using a one-way ANOVA and Bonferroni post hoc tests. The criterion for statistical significance was $p < 0.05$. Assumptions of normal distribution and homogeneity of variance were confirmed by Shapiro–Wilk and Levene tests, respectively. All data are presented as means ± standard errors of the mean (SEM) unless otherwise noted.

## Results

### Mechanical pain hypersensitivity after intraplantar incision

To investigate the role of LTB$_4$ in tissue injury-induced acute pain, we studied behavioral pain responses to a plantar incision in BLT1KO mice and wild-type littermates (BLT1WT). Withdrawal thresholds of the ipsilateral paw decreased in both BLT1KO and BLT1WT mice, and those of the ipsilateral paw were lowest at 2 hours, which was maintained for 7 days after the incision. BLT1KO mice exhibited reduced mechanical pain hypersensitivity as compared with BLT1WT mice in the ipsilateral paw on post-incisional days 2 ($p < 0.0001$), and day3, 4 ($p < 0.05$). The 50% paw withdrawal thresholds of the contralateral paw were not affected by the incision (Fig 1).

### Quantification of LTB$_4$ concentration in the plantar tissue

We next quantified the level of LTB$_4$ in the plantar tissue samples collected at different time points after the incision using LC-MS/MS. LTB$_4$ concentration peaked at 3hr and decreased, but moderate amounts of LTB$_4$ were also detected on days 1 and day 3 after incision. LTB$_4$ concentration increased significantly from baseline 3 hours and 1 day after the incision ($p < 0.001$, $p < 0.01$; Fig 2).

### Leukocyte infiltration in the peri-incisional tissue of the footpad

Because LTB$_4$ is a strong proinflammatory chemoattractant and is related to inflammatory conditions [1], we hypothesized that the infiltration of leukocytes in the peri-incisional site would be reduced in BLT1KO mice. To determine which profile of leukocyte populations in the peri-incision sites were affected by LTB$_4$-BLT1 signaling, we measured neutrophils (CD11b$^+$Ly6G$^+$) and non-neutrophil myeloid cells (CD11b$^+$Ly6G$^-$) using flow cytometry. Non-neutrophil myeloid cells were categorized according to differences in the expression of Ly6C as Ly6C$^{high}$ monocytes (inflammatory monocytes) or non-classic Ly6C$^{low}$ monocytes [3, 45–48].

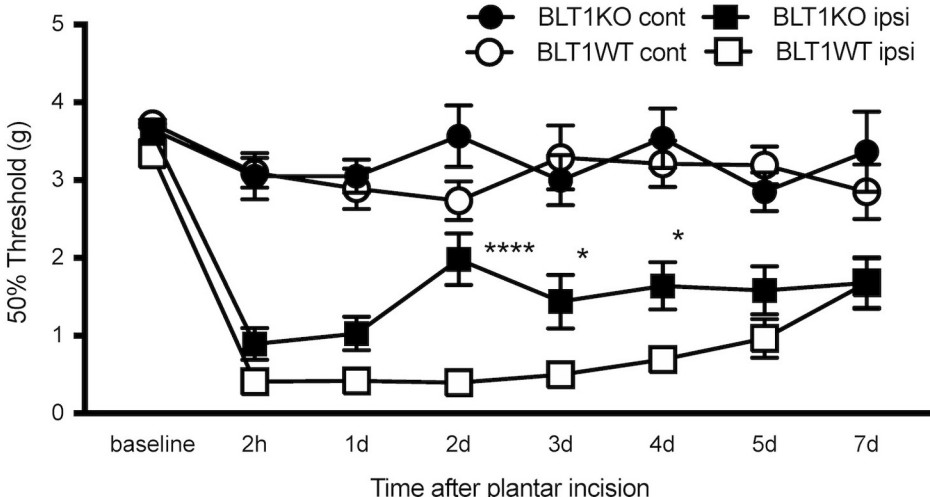

**Fig 1. Effect of BLT1 blockade on mechanical withdrawal threshold after plantar incision.** The 50% withdrawal thresholds in the ipsilateral and contralateral paws were measured at 2 h and 1, 2, 3, 4, 5, and 7 days after the plantar incision. The withdrawal thresholds of the ipsilateral paw decreased in both BLT1KO and BLT1WT mice. Data are shown as means ± SEM (n = 15–16 per group). A two-way ANOVA with Bonferroni post hoc tests was used for statistical analysis. *$p < 0.05$, ****$p < 0.0001$ BLT1KO ipsilateral vs. BLT1WT ipsilateral. The withdrawal thresholds of the contralateral paw were not affected by the plantar incision.

Infiltrated neutrophils peaked 1day after the incision, and there was no significant difference between BLT1WT and BLT1KO mice (Fig 3C). Infiltrated Ly6C$^{high}$ monocytes increased 1–3 days after the incision (Fig 3D). The accumulation of Ly6C$^{high}$ monocytes was significantly more attenuated in BLT1KO mice than in BLT1WT mice on post-incisional days 1 (p<0.05) and day3 (p<0.01). Ly6C$^{low}$ monocytes increased from day 3 and remained high until day 7 after the incision, but their levels did not differ significantly between BLT1WT and BLT1KO mice (Fig 3E).

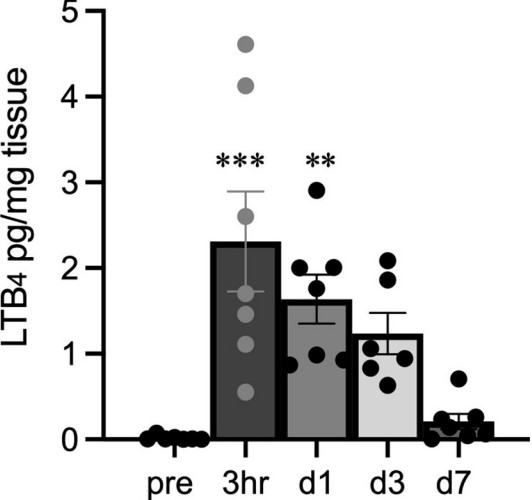

**Fig 2. Local LTB$_4$ concentration in the plantar tissue after the incision and pain assessment after local injection of LTB$_4$ in BLT1WT mice.** Quantitative analysis of LTB$_4$ concentration at the incisional site in BLT1WT mice was performed using LC-MS/MS. Data are expressed as pg of LTB$_4$ per mg of tissue and shown as means ± SEM (n = 6–7 per group). LTB$_4$ concentration increased significantly from baseline 3 hours and 1day after the incision. A one-way ANOVA with Bonferroni post hoc tests was used for statistical analysis. ***$p < 0.001$, **$p < 0.01$ vs. pre-incision.

## Cytokine analysis

Because cytokines produced in incised tissue are one of the primary mediators of the acute inflammatory response [32], we focused on proinflammatory cytokines in the incised tissue and compared their levels between BLT1KO and BLT1WT mice.

In BLT1WT mice, IL-1β and TNF-α levels increased from 3 hours after the incision and peaked 1 day after the incision. In BLT1KO mice, IL-1β and TNF-α levels were significantly lower than those in BLT1WT mice at day1 (p<0.0001) (Fig 4A and 4B). In contrast, IL-6 peaked 3 hours after the incision, and there was no significant difference between BLT1WT and BLT1KO mice (Fig 4C).

## Painful reactions following LTB$_4$ administration

To confirm the direct effect of LTB$_4$ increase to the pain responses, the experiments of LTB$_4$ injection and behavior tests were performed. Ten ng of LTB$_4$ injection into the planter surface of hind paw induced significant increase of guarding pain score compared with the injection of vehicle or injection of 1ng LTB$_4$. ($p < 0.01$, $p < 0.01$; Fig 5A). The 50% paw withdrawal threshold were significantly decreased at 20 min and 40 min following LTB$_4$ injection compared with vehicle injection ($p < 0.001$, $p < 0.01$; Fig 5B) and contralateral injection ($p < 0.001$; Fig 5B). LTB$_4$ might directly affect peripheral nociceptor and induce pain-related reaction transiently. Cytokine analysis in the paw at 3hr, 1day and 3day after LTB$_4$ injection were also performed. IL-1β, TNF-α and IL-6 elevations were small and not significantly different from the vehicle injection group (data not shown).

## Discussion

In the present study, we showed that BLT1KO mice developed attenuated pain behavior in the incisional pain model. We also observed a significant increase in endogenous LTB$_4$ in the tissue at the incision site during a relatively early phase following the plantar incision. Compared with wild-type mice, BLT1KO mice showed greater reductions in infiltrated inflammatory monocytes and inflammatory cytokines at the incision site, which suggested that LTB$_4$-BLT1 signaling in monocytes is a key element of persistent pain responses following tissue injury.

Flow cytometrical analysis of local infiltrating leukocytes in the peri-incisional area revealed a marked increase in neutrophils and inflammatory monocytes during the early phase and an increase in resident monocytes/macrophages during the late phase. Although neutrophils were the earliest to show an increase, previous studies have shown that they are not primarily involved in hyperalgesia in the incision model [44, 49]. Several drugs and antibodies that inhibit the infiltration of neutrophils have been shown to attenuate mechanical responses in an incisional pain model; however, they also have an inhibitory effect on monocytes/macrophages [50]. Ghasemlou et al. reported that the genetic depletion of CD11b$^+$Ly6G$^-$ myeloid cells (primarily monocytes and macrophages) significantly attenuate the pain response in the incisional pain model and revealed that these non-neutrophilic CD11b-positive myeloid cells (monocytes and macrophages) significantly contribute to the pain response at the incisional site [44]. In the present study, inflammatory monocytes were significantly more reduced, in parallel with the attenuation of the pain response, in BLT1KO mice than in BLT1WT mice, which supports the previous report that the infiltration of monocytes is important in the maintenance of incisional pain. BLT1 is expressed in various types of inflammatory cells, and the expression of BLT1 has been confirmed in circulating C-C Motif Chemokine Receptor 2 (CCR2)-high inflammatory monocytes and CCR2-low resident monocytes in peripheral blood [51]. Therefore, we suggest that LTB$_4$-BLT1 signaling contributed to the recruitment of inflammatory monocytes following the incision and the resulting pain response.

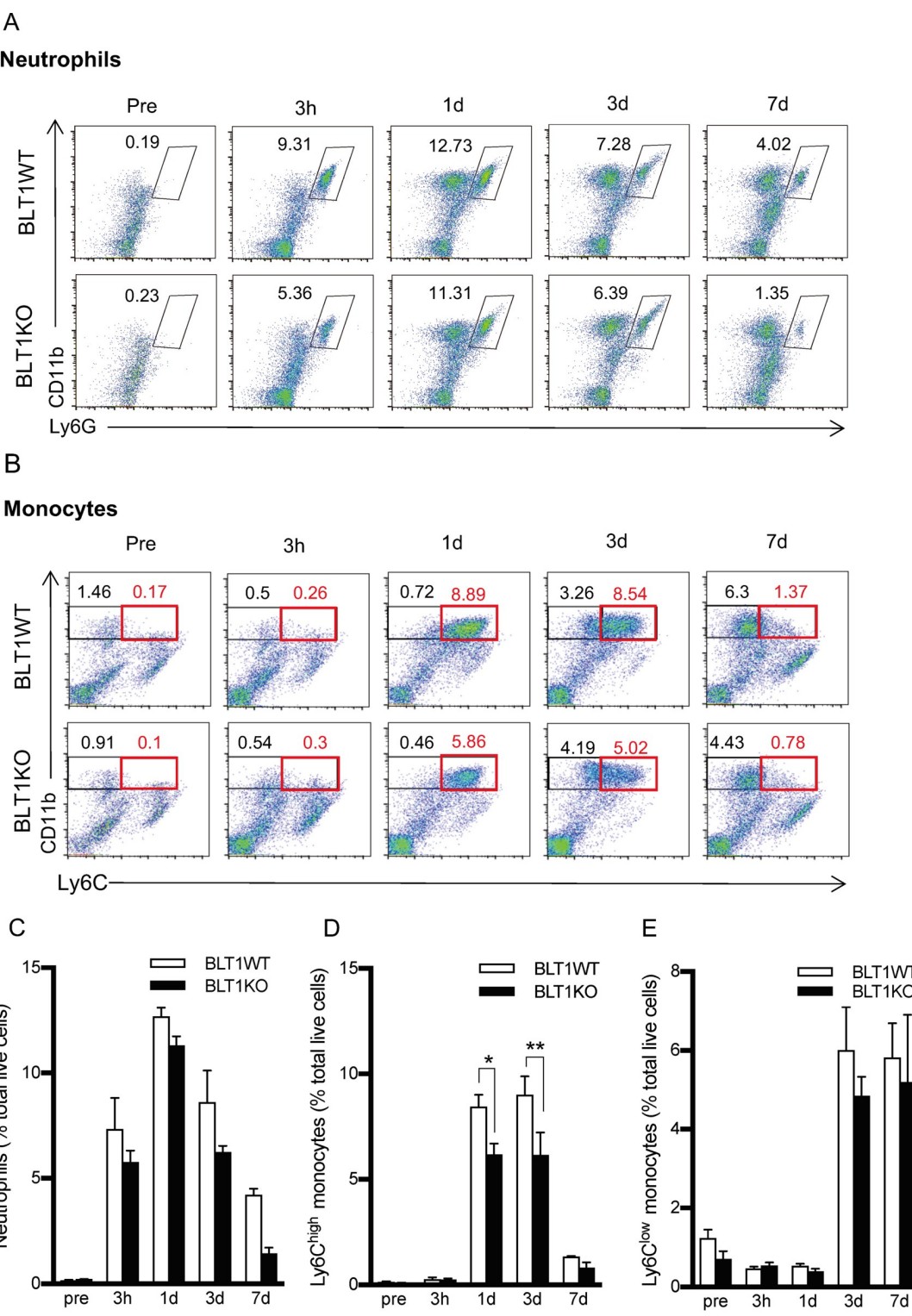

**Fig 3. Leukocyte infiltration in the plantar tissue after the paw incision.** The percentage of neutrophils (CD11b$^+$Ly6G$^+$) and Ly6C$^{high}$ and Ly6C$^{low}$ monocytes were measured in the plantar tissue collected from BLT1KO and BLT1WT mice at different time points after the incision. A, Representative forward/scatter dot plots showing the percentage of neutrophils. B, Representative forward/scatter dot plots showing percentages of Ly6C$^{high}$ (red square gate) and Ly6C$^{low}$ monocyte populations (black square gate). C, The percentage of neutrophils increased and peaked on day 1 in both groups, but there was no significant

difference between groups. D, The percentage of Ly6C$^{high}$ monocytes increased from days 1 to 3 after the incision in both groups. The percentage of Ly6C$^{high}$ monocytes in BLT1KO mice were more attenuated than that of BLT1WT mice on days 1 and 3 after the incision ($^*p < 0.05$, $^{**}p < 0.01$ vs. BLT1WT). E, The percentage of Ly6C$^{low}$ monocytes increased from day 3 after the incision and remained high until day 7 after the incision in both groups. However, there were no significant differences between groups. Data are shown as means ± SEM (n = 5–6 per group). A two-way ANOVA with Bonferroni post hoc tests was used for statistical analysis.

In our previous report, BLT1KO mice showed significantly reduced neutrophil-derived inflammatory responses, such as edema formation and Myeloperoxidase (MPO) activity, in the footpad induced by intraplantar formalin injection. These results were observed within 1 hour of the formalin injection and revealed that LTB₄-BLT1 signaling is responsible for neutrophil-derived inflammatory responses during the early phase. Given that the incisional pain model is considerably more invasive and persistent than the formalin-induced pain model, we speculate that not only the initial neutrophil infiltration, but also inflammatory monocytes infiltration induced by LTB₄-BLT1 signaling are involved in the pain enhancement observed in the incisional pain model. In contrast, Ghasemlou et al. also reported that inflammatory (CCR$^{2+}$Ly6C$^{high}$) monocytes were not responsible for mechanical pain hypersensitivity during incisional pain. In this study, the increase in the Ly6C$^{high}$ cell subset lasted longer than they reported, suggesting that intensity of inflammation may stronger than their model and the effect of Ly6C$^{high}$ cell subset appeared more potently. Emerging studies have shown that inflammatory monocytes (Ly6C$^{high}$ subset) play important roles in a various disease based on tissue damage, including liver injury [52], myocardial infarction [53], and skin injury [54]. In this study, reduction of inflammatory monocytes (Ly6C$^{high}$ subset) is almost parallel to the inhibition of pain-related responses in BLT1KO mice. A BLT1 antagonist may be effective as an anti-inflammatory analgesic secondary to NSAIDs, with fewer side effects during the early phase of postoperative pain. Additionally, treatments targeting LTB₄-BLT1 signaling in inflammatory monocytes (the Ly6C$^{high}$ subset) may be useful for decreasing inflammation to inhibit pain.

Infiltrated neutrophils and monocytes/macrophages are both responsible for the release of inflammatory mediators. Inflammatory mediators can directly activate and sensitize peripheral nociceptors. In particular, IL-1β has been shown to be strongly associated with the

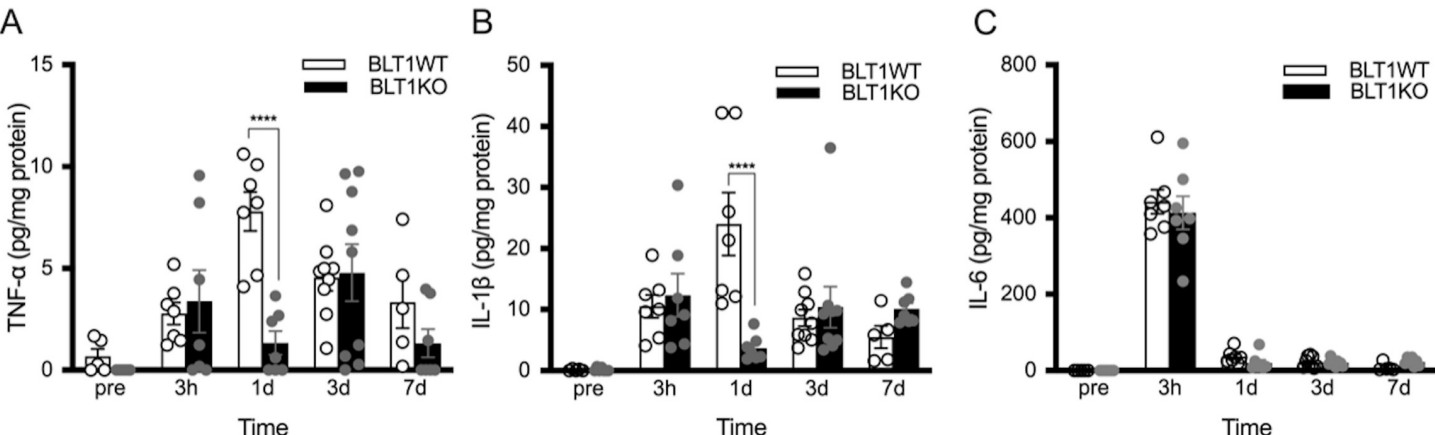

Fig 4. Level of inflammatory cytokines after plantar incision by time course. Cytokine levels in the planter tissue at pre-incision, 3 hours, and 1 and 7 days after the paw incision are shown. A, IL-1β. B, TNF-α. C, IL-6. In BLT1KO mice, IL-1β and TNF-α levels 1 day after the incision were significantly lower than those in BLT1WT mice. No significant difference in IL-6 levels was found between the groups. Data are shown as means ± SEM (n = 5–7 per group). A two-way ANOVA with Bonferroni post hoc tests was used for statistical analysis. $^{****}p < 0.0001$ vs. BLT1WT mice.

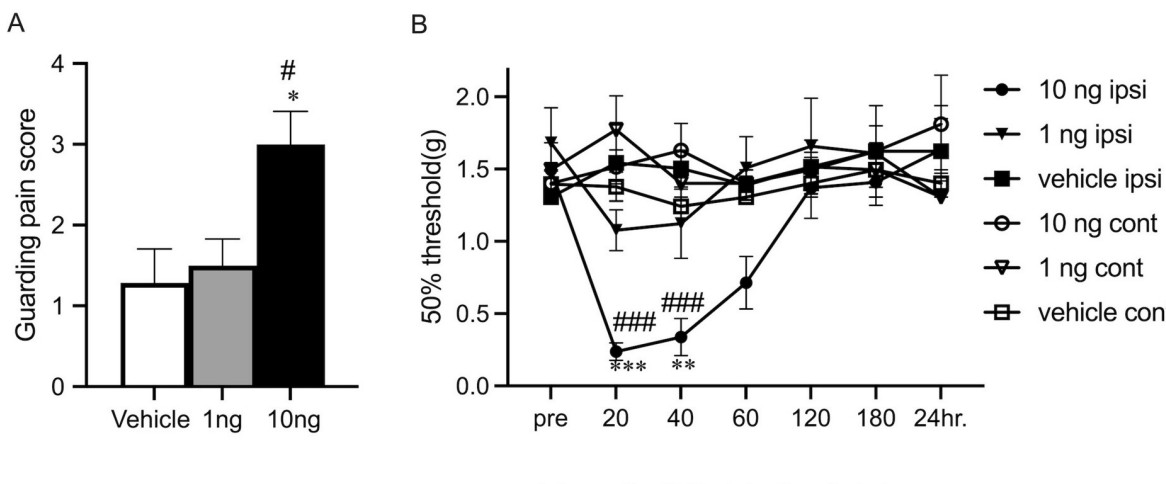

**Fig 5. Pain assessment after local injection of LTB$_4$ in BLT1WT mice.** A, Effect of injection of LTB$_4$ on Guarding behavior. Guarding pain behavior measured during a 30 min period after injection. The results are presented as mean ± SEM (n = 6–7 per group) One-way ANOVA, with Bonferroni post hoc tests was used for statistical analysis. $^*p < 0.01$ vs. vehicle-injection and # $p < 0.01$ vs. 1ng LTB$_4$ injection. B, Effect of injection of LTB$_4$ on mechanical pain thresholds. The results are presented as mean ± SEM (n = 7 per group). A two-way ANOVA with Bonferroni post hoc tests was used for statistical analysis. $^{***}p < 0.001$, $^{**}p < 0.01$ vs. vehicle-incision and ### $p < 0.001$ vs. contralateral injection.

modulation of acute pain during the early phase following tissue injury [32, 34, 55]; moreover, TNF-α is known to induce peripheral nociceptor sensitization [56]. We confirmed that, in the injured site 24 h after the incision, the increase in IL-1β and TNF-α was more reduced in BLT1KO mice than in BLT1WT mice. Cell subsets infiltrating the paw suggested that cytokines elevated at day1 are primarily produced from inflammatory monocytes. Swirski et al. presented that inflammatory monocyte (Ly6C$^{high}$ subset) are more enriched in MPO activity than nonclassical monocytes (Ly6C$^{low}$ subset) and this MPO-rich inflammatory monocytes (Ly6C$^{high}$ subset) promote inflammation and tissue destruction through the release of proteases and inflammatory cytokines such as TNF-α in various inflammatory settings including infection and injury [57]. Since the cytokine elevation was transient, it remains to be elucidated how these cytokines up-regulation might be involved in the persistence of postoperative pain.

LTB$_4$ injection induced a transient spontaneous pain response and lowered the mechanical threshold until 40 minutes after injection. These results were consistent with the previous report of rat [20]. It is speculated that peripherally increased LTB$_4$ induce pain-related responses via Transient Receptor Potential Vanilloid 1(TRPV1) expressed in the afferent sensory nerves of the skin. Several reports have shown that LTB$_4$ sensitizes primary afferent nociceptors directly without the action of its specific receptor BLT1 [58, 59]. In our model, LTB$_4$ production at the injured site was transient, and it was difficult to determine whether the direct sensitization of the peripheral nerve by LTB$_4$ was persistent and induced prolonged incisional pain. Moreover, LTB$_4$ and the lipid product of lipoxygenase are considered one candidate for the endogenous agonist of the Transient Receptor Potential Vanilloid 1(TRPV1) receptor at the peripheral terminal [60–62]. However, an extremely high concentration of LTB$_4$ (EC50 = 11.7 μM) is required to sensitize TRPV1 [60]. By contrast, Zinn et al. reported that relatively low concentrations of LTB$_4$ (100–200 nM) enhance TRPV1-mediating calcium increases in a BLT1-dependent manner by using the primary culture of the dorsal root ganglion, which suggests that BLT1 is involved in TRPV1 activation [58]. However, it was also difficult to determine whether the transiently increased local LTB$_4$ induced persistent TRPV1

sensitization and maintained incisional pain. Another group reported that high concentration of LTB4 (10 μM) induced calcium increase in primary culture of the dorsal root ganglion [21]. It is unclear how peripherally produced LTB$_4$ acts its receptor BLT1 expressed in the dorsal root ganglion and mediate pain sensitization.

There is also a possibility that LTB$_4$-BLT1 signaling in the spinal cord was involved in the pain response after the incision. Spinal glial cells, such as astrocytes and microglia, have been reported to contribute to mechanical pain hypersensitivity in the incisional model [63] and are strong candidates for the source of LTB$_4$. LTB$_4$ released from glial cells binds to its high-affinity receptor BLT1 in the neuron to accelerate neuronal activity in the spinal cord. Further analysis is required to clarify whether glial cells release LTB$_4$ and whether LTB$_4$-BLT1 signaling in the central terminal is involved in the mechanism underlying incisional pain.

## Conclusions

In a post-incisional pain model, we found that the blockade of LTB$_4$-BLT1 signaling following paw incision attenuated mechanical pain hypersensitivity, reduced local accumulation of inflammatory monocytes, and suppressed the local increase in IL-1β and TNF-α. These results suggest that the LTB$_4$-BLT1 axis is involved in peripheral sensitization by promoting the recruitment of inflammatory monocytes in the inflamed site and is a promising novel therapeutic target for post-operative pain.

## Acknowledgments

We would like to thank Ms. Tokie Totsu (Department of Anesthesiology, Faculty of Medicine, The University of Tokyo) and Ms. Mai Ohba (Department of Biochemistry, Juntendo University School of Medicine) for experimental support.

## Author Contributions

**Conceptualization:** Nobuko Ito.

**Data curation:** Miho Asahara, Nobuko Ito, Yoko Hoshino.

**Formal analysis:** Miho Asahara, Nobuko Ito, Takaharu Sasaki, Motonao Nakamura.

**Funding acquisition:** Nobuko Ito.

**Investigation:** Miho Asahara, Nobuko Ito, Yoko Hoshino, Motonao Nakamura.

**Methodology:** Nobuko Ito.

**Resources:** Yoshitsugu Yamada.

**Supervision:** Takehiko Yokomizo, Takao Shimizu, Yoshitsugu Yamada.

**Visualization:** Nobuko Ito.

**Writing – original draft:** Miho Asahara, Nobuko Ito.

**Writing – review & editing:** Nobuko Ito, Takehiko Yokomizo, Motonao Nakamura, Takao Shimizu.

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
