## [Decision Letter · Decision Letter 0]

1 May 2022

PONE-D-22-08079Role of leukotriene B4 (LTB4)-LTB4 receptor 1 signaling in post-incisional nociceptive sensitization and local inflammationPLOS ONE

Dear Dr. Ito,

Thank you for submitting your manuscript to PLOS ONE. After careful consideration, we feel that it has merit but does not fully meet PLOS ONE’s publication criteria as it currently stands. Therefore, we invite you to submit a revised version of the manuscript that addresses the points raised during the review process. Please take into consideration the comments made by Reviewer 1 and 2 regarding the discrepancies between your data and data in the literature. The mechanisms by which BLT1 is mediating Incisional pain need to be further addressed specially the role of different pain pathway compartments.  

We look forward to receiving your revised manuscript.

Kind regards,

Thiago Mattar Cunha, PhD

Academic Editor

PLOS ONE

Journal Requirements:

Reviewers' comments:

Reviewer's Responses to Questions

**Comments to the Author**

1. Is the manuscript technically sound, and do the data support the conclusions?

Reviewer #1: Partly

Reviewer #2: Partly

Reviewer #3: Yes

2. Has the statistical analysis been performed appropriately and rigorously? 

Reviewer #1: Yes

Reviewer #2: No

Reviewer #3: Yes

3. Have the authors made all data underlying the findings in their manuscript fully available?

Reviewer #1: No

Reviewer #2: Yes

Reviewer #3: Yes

4. Is the manuscript presented in an intelligible fashion and written in standard English?

Reviewer #1: Yes

Reviewer #2: Yes

Reviewer #3: Yes

5. Review Comments to the Author

Reviewer #1: The manuscript by Asahara investigates the contribution of BLT1 in the development of mechanical pain hypersensitivity produced by incision and the related inflammation. The authors show that Blt1 knock-out animals express reduced pain hypersensitivity after incision compared to wild-type animals. They also report that incision produces, in a time-dependent manner, an increase in LTB4 in the plantar tissue. The genetic deletion of Blt1 reduces the infiltration of monocytes and the production of IL-1beta and TNFalpha in the plantar tissue.

Overall, the topic addressing the mechanisms responsible for surgical pain is interesting. The results are clearly presented and the manuscript easy to read. An important issue here is the mechanisms of action of BLT1 that is not clearly investigated making the reading of the manuscript quite confusing. Indeed, the authors suggest a local mechanisms involving BLT1 expressed in immune cells promoting the recruitment of monocytes. However, BLT-1 is also express in sensory neurons of the DRG and the spinal cord, as reported by the authors. Hence, the experiments and the data presented in the manuscript do not allow to conclude on how BLT1 participates in the development of incision-induced pain hypersensitivity. Extensive revision is needed to address such an important issue. For instance, additional experiments evaluating the effects of intraplantar LTB4 injection on pain sensitivity, monocytes infiltration and cytokines production could be performed.

The additional comments should be considered by the authors.

- Additional experiments with the use of a non-reflexive test should be conducted. The sole description of pain-related behaviors via the use of reflexive test is insufficient to conclude on alteration of pain behaviors. For instance, pain score, place preference paradigm, catwalk etc…are methods used to assess pain-related behaviors without the use of a reflexive response.

- The effects of Blt1 deletion seems moderate and transient on the mechanical pain hypersensitivity. One may question on the importance of Blt1 in incision-induced pain-related behaviors, especially from a clinical point of view. The authors should comment.

- To exclude potential compensatory mechanisms in Blt1 Knock-out mice, pharmacological experiments targetting BLT1 should be considered by the authors. For instance, the effects of intraplantar injection of the BLT1 antagonist ONO-4057 could be evaluated in the incisional pain model.

- Have the experiments been performed in a blinded manner ? This should be indicated in the manuscript. If not, the authors should justify.

- It is admitted that the use of the up and down method for the the von frey test allows to evaluate not hyperalgesia but allodynia. Actually, for the description of the results the terms "Mechanical hyperalgesia" should be changed by "mechanical pain hypersensitivity" as the reflexive test does not allow to truly assess hyperalgesia/allodynia in animals.

- Plantar tissue has been collected to perform biochemical and cellular analysis. However, it is unclear what represents this tissue. Does this include muscle and skin or just the plantar muscle? The authors should comment.

Reviewer #2: Miho Asahara and colleagues evaluated LTB4/BLT1 in post-incisional pain.

Major points

The role of BLT1 was evident at days 2, 3 and 4 post-incision (Figure 1). Why LTB4 production was quantitated at 3h and days 1 and 7? Using the same tool, the authors can ascertain the LTB4 production at time points that were observed as important since BLT1 deficiency caused a reduction of mechanical allodynia at days 2, 3 and 4 post-incision (Figure 1). It is possible that the role of LTB4/BLT1 has consequences that goes beyond the time point of LTB4 production. However, the lack of data at those days do not allow reaching a conclusion. Overall, there seems to be an arbitrary selection of time points of analyzes that prevent putting all data in line to evaluate them. Figure 1 checked many time points, figures 2 and 4 verified 3h, 1 and 7 days, and figure 3 verified 3h, 1, 3 and 7 days.

Why LTB4 production was evident at 3h and then significant alteration of pain was observed only at 2, 3 and 4 days later? Note that this is a difference of many days. It would be interesting to check if injection of LTB4 produces a similar profile.

Quite interestingly, inflammatory monocytes were increased at days 1 and 3, and their recruitment was inhibited by BLT1 deficiency. These results suggest a parallel between pain and inflammatory monocyte recruitment. However, the data do not demonstrate whether at all time points with difference in phenotype there is also difference in LTB4 production. This is important to be done and further supports the question above.

Furthermore, which is the cellular target of LTB4? Do Ly6Chigh inflammatory monocytes express BLT1? Is it a direct effect (or lack of effect in the case of BLT1 deficient mouse) of LTB4 in BLT1 expressed by Ly6Chigh inflammatory monocytes? Are Ly6Chigh inflammatory monocytes producing the cytokines?

The discrepancy between the present finding and the previous finding (DOI: 10.1073/pnas.1501372112) should be better explored with experiments. These papers are suggesting different cellular populations are involved in incisional pain. What are Ly6Chigh inflammatory monocytes doing in incisional pain and how LTB4 can affect their function? What is the explanation for the difference at day 1? At this time point there is no significant analgesia by BLT1 deficiency, but there is reduction of Ly6Chigh inflammatory monocytes and IL-1b and TNFa production. However, Ly6Chigh inflammatory monocytes are high at day 3 and we do not know the cytokine profile.

The current data do not allow to conclude if the phenotype of BLT1 deficiency is a result of direct function of LTB4 in neurons or non-neuronal cells or if it depends on indirect effect of tissue resident cells, which produce other molecules that are the actual actors to produce pain and inflammatory cell recruitment or both. LTB4 can exert neuronal and non-neuronal effects that will lead to pain. The neuronal mechanism (DOI: 10.1016/j.molbrainres.2005.02.029) was clearly not investigated in the present study and the authors used a model of pain.

What is the level of LTB4 in the spinal cord? These authors have shown the increase of pCREB in the spinal cord in the formalin test is BLT1 dependent. What happens in the incisional pain model?

In a prior study, the authors used the MPO assay (doi: 10.1186/s12990-015-0010-9). To give a more comparative view of the results and align them with the literature, it is important to have MPO data and show which cells are producing MPO in the present study. Do these cells express the BLT1 receptors? Do inflammatory monocytes (Ly6Chigh) produce MPO in the present experimental condition? Please, see (doi: 10.1172/JCI42304).

Figure 2. The number of samples ranged from 4 to 7. It is a huge variation in the number of samples. Why?

A huge variation also occurred in Figure 4 with n varying from 5 to 9. Why?

There is no mention on repetition of experiments.

There is no mention about data normality and homogeneity.

Figures 1 and 2 can be combined.

Reviewer #3: In this study, the authors have shown that plantar incision-induced mechanical hypersensitivity, accumulation of inflammatory monocytes and the production of IL-1β and TNF-α in the incised area were significantly reduced in BLT1-knockout mice. Although the present results can be partly expected from previous findings, this study provides further insights into the roles of LTB4-BLT1 signaling in postoperative incisional pain. However, there are some points that should be addressed by the authors, as described below.

1) To indicate that this study is basic research using experimental animals, the animal species should be included in the title.

2) The basal 50% paw withdrawal threshold in C57BL/6 mice is usually less than 1 g. Have you made any mistakes in the calculations in up-down method of von Frey filament test? Provide the detailed methods in the test and the calculated method.

3) Significant reduction of mechanical hypersensitivity in BLT1KO mice was still not observed at 2h and 1d, but peaked between 2d and 4d after the plantar incision. By contrast, the accumulation of inflammatory monocytes and the production of IL-1β and TNF-α were already reduced at 1 day after the plantar incision. How do the authors explain this time gap? The authors should discuss this point in Discussion section.

6. PLOS authors have the option to publish the peer review history of their article (what does this mean?). If published, this will include your full peer review and any attached files.

Reviewer #1: No

Reviewer #2: No

Reviewer #3: No

---

## [Author Response · Author response to Decision Letter 0]

14 Jul 2022

We appreciate all reviewers for reviewing our manuscript and giving useful comments to improve our manuscript. 

Reviewer #1: The manuscript by Asahara investigates the contribution of BLT1 in the development of mechanical pain hypersensitivity produced by incision and the related inflammation. The authors show that Blt1 knock-out animals express reduced pain hypersensitivity after incision compared to wild-type animals. They also report that incision produces, in a time-dependent manner, an increase in LTB4 in the plantar tissue. The genetic deletion of Blt1 reduces the infiltration of monocytes and the production of IL-1beta and TNFalpha in the plantar tissue.

Overall, the topic addressing the mechanisms responsible for surgical pain is interesting. The results are clearly presented and the manuscript easy to read. An important issue here is the mechanisms of action of BLT1 that is not clearly investigated making the reading of the manuscript quite confusing. Indeed, the authors suggest a local mechanisms involving BLT1 expressed in immune cells promoting the recruitment of monocytes. However, BLT-1 is also express in sensory neurons of the DRG and the spinal cord, as reported by the authors. Hence, the experiments and the data presented in the manuscript do not allow to conclude on how BLT1 participates in the development of incision-induced pain hypersensitivity. Extensive revision is needed to address such an important issue. For instance, additional experiments evaluating the effects of intraplantar LTB4 injection on pain sensitivity, monocytes infiltration and cytokines production could be performed.

We evaluated the effects of LTB4 injection on pain sensitivity (Fig. 5), and we observed transient pain responses. We speculated that peripherally-increased LTB4 induces pain-related responses via both the BLT1 pathway, expressed in inflammatory monocytes, and the TRPV1 pathway, expressed in the afferent sensory nerves of the skin. However, it was unclear whether LTB4 directly acted on TRPV1 because LTB4 disappeared rapidly. We believe that the direct effect of LTB4 on the peripheral nerve endings in the skin will not be long-lasting. Based on our results, we propose that LTB4-induced inflammation, mainly the recruitment of inflammatory monocytes, is more strongly involved than TRPV1 activation of peripheral nerves by LTB4 in the development of incision-induced hypersensitivity.

The additional comments should be considered by the authors.

- Additional experiments with the use of a non-reflexive test should be conducted. The sole description of pain-related behaviors via the use of reflexive test is insufficient to conclude on alteration of pain behaviors. For instance, pain score, place preference paradigm, catwalk etc…are methods used to assess pain-related behaviors without the use of a reflexive response.

Unfortunately, because of the shortage of mice, we were unable to perform the non-reflexive test to compare the behaviors of BLT1KO and BLT1WT mice. As suggested by the reviewers, we have evaluated the effects of intraplantar LTB4 injection on pain sensitivity using the guarding pain score, which enabled us to assess spontaneous pain-related behaviors. (Fig. 5)

- The effects of Blt1 deletion seems moderate and transient on the mechanical pain hypersensitivity. One may question on the importance of Blt1 in incision-induced pain-related behaviors, especially from a clinical point of view. The authors should comment.

Although NSAIDs are standard analgesics administered for postoperative pain, their use can present challenges because of side effects, such as aspirin-induced asthma and gastrointestinal damage. A BLT1 antagonist may be effective as an anti-inflammatory analgesic secondary to NSAIDs, with fewer side effects during the early phase of postoperative pain. Additionally, treatments targeting LTB4-BLT1 signaling in inflammatory monocytes (the Ly6Chigh subset) may be useful for decreasing inflammation to inhibit pain. (L346-353)

- To exclude potential compensatory mechanisms in Blt1 Knock-out mice, pharmacological experiments targetting BLT1 should be considered by the authors. For instance, the effects of intraplantar injection of the BLT1 antagonist ONO-4057 could be evaluated in the incisional pain model.

We could not obtain a sufficient amount of the BLT1 antagonist to perform the experiments because ONO-4057 is not commercially available. However, we hypothesize that intraplantar administration of a BLT1 antagonist would attenuate incisional pain because painful behavior was observed after intraplantar injection of LTB4. Moreover, local BLT1 antagonist administration may have an inhibitory effect on inflammatory cell infiltration and cytokine increase.

- Have the experiments been performed in a blinded manner ? This should be indicated in the manuscript. If not, the authors should justify.

The experiments were performed in a blinded manner. We have described this in the text. (L130 and142)

- It is admitted that the use of the up and down method for the the von frey test allows to evaluate not hyperalgesia but allodynia. Actually, for the description of the results the terms "Mechanical hyperalgesia" should be changed by "mechanical pain hypersensitivity" as the reflexive test does not allow to truly assess hyperalgesia/allodynia in animals.

We changed the term “Mechanical hyperalgesia” by “Mechanical pain hypersensitivity”.

- Plantar tissue has been collected to perform biochemical and cellular analysis. However, it is unclear what represents this tissue. Does this include muscle and skin or just the plantar muscle? The authors should comment.

Description of “Plantar tissue” was added to the method section, as “Plantar tissue, including skin and underlying muscle.” 

Reviewer #2: Miho Asahara and colleagues evaluated LTB4/BLT1 in post-incisional pain.

Major points

The role of BLT1 was evident at days 2, 3 and 4 post-incision (Figure 1). Why LTB4 production was quantitated at 3h and days 1 and 7? Using the same tool, the authors can ascertain the LTB4 production at time points that were observed as important since BLT1 deficiency caused a reduction of mechanical allodynia at days 2, 3 and 4 post-incision (Figure 1). It is possible that the role of LTB4/BLT1 has consequences that goes beyond the time point of LTB4 production. However, the lack of data at those days do not allow reaching a conclusion. Overall, there seems to be an arbitrary selection of time points of analyzes that prevent putting all data in line to evaluate them. Figure 1 checked many time points, figures 2 and 4 verified 3h, 1 and 7 days, and figure 3 verified 3h, 1, 3 and 7 days.

Why LTB4 production was evident at 3h and then significant alteration of pain was observed only at 2, 3 and 4 days later? Note that this is a difference of many days. It would be interesting to check if injection of LTB4 produces a similar profile.

Additional experiments on the quantification of LTB4 concentration revealed that LTB4 was detected on days 1 and 3. As suggested by several reviewers, we have evaluated the effects of LTB4 injection on pain sensitivity, and we observed transient pain responses. We speculate that peripherally-increased LTB4 induces pain-related responses via both the BLT1 pathway, expressed in inflammatory monocytes, and/or the TRPV1 pathway, expressed in the afferent sensory nerves of the skin.

Quite interestingly, inflammatory monocytes were increased at days 1 and 3, and their recruitment was inhibited by BLT1 deficiency. These results suggest a parallel between pain and inflammatory monocyte recruitment. However, the data do not demonstrate whether at all time points with difference in phenotype there is also difference in LTB4 production. This is important to be done and further supports the question above.

We quantified LTB4 concentration in the plantar tissue on days 1 and 3 after the incision, the results of which are provided in the new Fig. 2. LTB4 concentration peaked at 3 h, and a moderate level of LTB4 was detected not only on day 1 but also on day 3, which indicated that the LTB4 increase affected Ly6Chigh monocyte recruitment and strengthened inflammation on days 1 and 3. 

In regard to the reasons why the pain response in BLT1KO mice was not suppressed on day 1, we speculate that the inflammatory reaction was relatively strong on day 1, and stronger monocyte suppression was necessary for pain inhibition during this period.

Furthermore, which is the cellular target of LTB4? Do Ly6Chigh inflammatory monocytes express BLT1? Is it a direct effect (or lack of effect in the case of BLT1 deficient mouse) of LTB4 in BLT1 expressed by Ly6Chigh inflammatory monocytes? Are Ly6Chigh inflammatory monocytes producing the cytokines?

In general, BLT1 is highly expressed in neutrophils, and LTB4-BLT1 signaling in neutrophils is the primary initial inflammation response. Consequently, a cascade of the accumulation of neutrophils and other inflammatory cells is induced. In our previous report (Ref. 52), the expression of BLT1 was confirmed in CCR2-high inflammatory monocytes, which were identified as Ly6Chigh inflammatory monocytes. We propose that Ly6Chigh inflammatory monocytes are a target of LTB4 and that LTB4-BLT1 signaling in inflammatory monocytes plays an important role in the progression of inflammation. 

Furthermore, according to the reference mentioned below (doi: 10.1186/s12990-015-0010-9), MPO-rich Ly6Chigh monocytes promote inflammation and tissue destruction via the release of proteases and inflammatory cytokines, such as TNF-α, in various inflammatory settings.

The discrepancy between the present finding and the previous finding (DOI: 10.1073/pnas.1501372112) should be better explored with experiments. These papers are suggesting different cellular populations are involved in incisional pain. What are Ly6Chigh inflammatory monocytes doing in incisional pain and how LTB4 can affect their function? What is the explanation for the difference at day 1? At this time point there is no significant analgesia by BLT1 deficiency, but there is reduction of Ly6Chigh inflammatory monocytes and IL-1b and TNFa production. However, Ly6Chigh inflammatory monocytes are high at day 3 and we do not know the cytokine profile.

Previous reports from other group (DOI: 10.1073/pnas.1501372112) have shown that monocytes (nonneutrophil myeloid cells) are more involved than neutrophils in postoperative pain. In our study, we showed that inhibition of monocytes by suppressing BLT1 signaling also contributes to pain control. BLT1-deficiency resulted in the suppression of monocyte infiltration on days 1 and day 3, and may also be involved in the suppression of cytokines on day 1. 

Regarding the reason why the pain response in BLT1KO was not suppressed on day 1, we speculate that inflammatory reaction was relatively strong on day 1; thus, the stronger monocyte suppression may be necessary for pain inhibition.

The data of cytokine production at day 3 was added to the new Fig 4.

The current data do not allow to conclude if the phenotype of BLT1 deficiency is a result of direct function of LTB4 in neurons or non-neuronal cells or if it depends on indirect effect of tissue resident cells, which produce other molecules that are the actual actors to produce pain and inflammatory cell recruitment or both. LTB4 can exert neuronal and non-neuronal effects that will lead to pain. The neuronal mechanism (DOI: 10.1016/j.molbrainres.2005.02.029) was clearly not investigated in the present study and the authors used a model of pain.

We focused on the role of inflammatory cells and LTB4-BLT1 signaling in an incisional pain model. It remains unclear how peripherally-produced LTB4 acts on the DRG; however, it is possible that LTB4 activates BLT1 expressed in the small TRPV1-positive neurons of the DRG and mediates pain sensitization. A description of this has been added to the discussion section. (L369-373, L387-388)

What is the level of LTB4 in the spinal cord? These authors have shown the increase of pCREB in the spinal cord in the formalin test is BLT1 dependent. What happens in the incisional pain model?

In this study, the effect of LTB4-BLT1 signaling on central sensitization, such as an increase of pCREB in the dorsal horn of the spinal cord, was not confirmed. We focused on the peripheral mechanisms, determined which types of inflammatory cells are involved, and investigated whether LTB4 is produced at the site of injury and whether inflammatory cell infiltration is affected by the deficiency of the LTB4 receptor BLT1. LTB4 was not detected in the spinal cord on day 3 of the incisional pain model (data not shown). BLT1 expressed in the small neurons of the DRG may have impacted the modulation of incisional pain. 

In a prior study, the authors used the MPO assay (doi: 10.1186/s12990-015-0010-9). To give a more comparative view of the results and align them with the literature, it is important to have MPO data and show which cells are producing MPO in the present study. Do these cells express the BLT1 receptors? 

According to the single-cell RNASeq data (http://bis.zju.edu.cn/MCA/search2.html), cluster 17 with high expression of BLT1 is a neutrophil with high MPO and also expresses Ly6C1 and Ly6C2. In addition, cluster 5 macrophages express Ly6C2, BLT1, and MPO. These data indicated that neutrophils and monocytes/macrophages express both BLT1 and MPO, and release MPO via LTB4-BLT1 signaling. Although it was difficult in our study to determine whether the source of MPO is neutrophils or monocytes, MPO release may contribute to inflammation and pain enhancement.

 　　　　　　　　　　　　　　　　　 　　　　　　　　　　

　　　　　　　　　　　　　　　　　　　　　　　　　　　　　　　　　　　　　　　　　　　　　　　　　　　　　　　　　　　　　　　　

Do inflammatory monocytes (Ly6Chigh) produce MPO in the present experimental condition? Please, see (doi: 10.1172/JCI42304).

Thank you for the information about inflammatory monocytes (Ly6Chigh) and MPO production. The study reported that inflammatory monocytes (Ly6Chigh) express a 10-fold higher level of MPO than non-classical monocytes (Ly6Clow). We have described this in the discussion section. (L360-368) 

Figure 2. The number of samples ranged from 4 to 7. It is a huge variation in the number of samples. Why?

A huge variation also occurred in Figure 4 with n varying from 5 to 9. Why?

There is no mention on repetition of experiments.

There is no mention about data normality and homogeneity.

We quantified LTB4 concentration in the paw at the missing time points, and the sample ranges have been corrected. Individual data are shown in Fig. 2, which illustrates the variation of the data of each sample.

We apologize for the incorrect sample size provided in Fig. 4. We have corrected the sample size, and individual data are shown in the figure to illustrate the variation of the data of each sample.

Figures 1 and 2 can be combined.

Reviewer #3: In this study, the authors have shown that plantar incision-induced mechanical hypersensitivity, accumulation of inflammatory monocytes and the production of IL-1β and TNF-α in the incised area were significantly reduced in BLT1-knockout mice. Although the present results can be partly expected from previous findings, this study provides further insights into the roles of LTB4-BLT1 signaling in postoperative incisional pain. However, there are some points that should be addressed by the authors, as described below.

1) To indicate that this study is basic research using experimental animals, the animal species should be included in the title.

The animal species used in our study have been added to the title.

2) The basal 50% paw withdrawal threshold in C57BL/6 mice is usually less than 1 g. Have you made any mistakes in the calculations in up-down method of von Frey filament test? Provide the detailed methods in the test and the calculated method.

The details of the methods of assessment and calculation of the mechanical responses are described in the method section. (L104-121)

3) Significant reduction of mechanical hypersensitivity in BLT1KO mice was still not observed at 2h and 1d, but peaked between 2d and 4d after the plantar incision. By contrast, the accumulation of inflammatory monocytes and the production of IL-1β and TNF-α were already reduced at 1 day after the plantar incision. How do the authors explain this time gap? The authors should discuss this point in Discussion section.

We quantified LTB4 concentration in the plantar tissue on days 1 and 3 after the incision, and the results have been added to the new Fig. 2. LTB4 concentration peaked at 3 h and was still detectable on day 3, which indicated that a persistent LTB4 increase affects Ly6Chigh monocyte recruitment and strengthens inflammation on days 1 and 3. BLT1-deficiency suppressed monocyte infiltration on days 1 and 3, and may also be involved in the suppression of cytokines on day 1.

Regarding the reasons why the pain response in KO mice was not suppressed on day 1, we speculate that the inflammatory reaction was relatively strong on day 1 and, thus, stronger monocyte suppression may have been necessary for pain inhibition during this period.

---

## [Decision Letter · Decision Letter 1]

3 Aug 2022

PONE-D-22-08079R1Role of leukotriene B4 (LTB4)-LTB4 receptor 1 signaling in post-incisional nociceptive sensitization and local inflammation in micePLOS ONE

Dear Dr. Ito,

Thank you for submitting your manuscript to PLOS ONE. After careful consideration, we feel that it has merit but does not fully meet PLOS ONE’s publication criteria as it currently stands.Therefore, we invite you to submit a revised version of the manuscript that addresses the points raised during the review process.

Because of the unavailability of the original editor, I am handling your submission.

Your revised manuscript has been evaluated by three reviewers. Though two of them are more positive, Reviewer #2 raised serious concerns. In particular, as the reviewer pointed out, the results reported in 10.1073/pnas.1501372112 do not support the notion that CCR2+Ly6Chi cells contribute to pain. Accordingly, the mechanistic role of Ly6Chi monocytes in pain in your model needs to be further addressed, and your findings need to be better correlated with the literature.

We look forward to receiving your revised manuscript.

Kind regards,

Ichiro Manabe

Academic Editor

PLOS ONE

Additional Editor Comment:

Because of the unavailability of the original editor, I am handling your submission.

Your revised manuscript has been evaluated by three reviewers. Though two of them are more positive, Reviewer #2 raised serious concerns. In particular, as the reviewer pointed out, the results reported in 10.1073/pnas.1501372112 do not support the notion that CCR2+Ly6Chi cells contribute to pain. Accordingly, the mechanistic role of Ly6Chi monocytes in pain in your model needs to be further addressed, and your findings need to be better correlated with the literature.

Reviewers' comments:

Reviewer's Responses to Questions

**Comments to the Author**

1. If the authors have adequately addressed your comments raised in a previous round of review and you feel that this manuscript is now acceptable for publication, you may indicate that here to bypass the “Comments to the Author” section, enter your conflict of interest statement in the “Confidential to Editor” section, and submit your "Accept" recommendation.

Reviewer #1: (No Response)

Reviewer #2: (No Response)

Reviewer #3: All comments have been addressed

2. Is the manuscript technically sound, and do the data support the conclusions?

Reviewer #1: Yes

Reviewer #2: No

Reviewer #3: Yes

3. Has the statistical analysis been performed appropriately and rigorously? 

Reviewer #1: Yes

Reviewer #2: No

Reviewer #3: Yes

4. Have the authors made all data underlying the findings in their manuscript fully available?

Reviewer #1: (No Response)

Reviewer #2: Yes

Reviewer #3: Yes

5. Is the manuscript presented in an intelligible fashion and written in standard English?

Reviewer #1: Yes

Reviewer #2: No

Reviewer #3: Yes

6. Review Comments to the Author

Reviewer #1: The authors did an excellent work in addressing my comments and those raised by the different reviewers. The changes made by the authors greatly improve the manuscript. The remaining comments should be considered by the authors

Regarding the LTB4 plantar injection, a suggestion would be to evaluate cytokine concentration in plantar tissue to support the peripheral action of LTB4 on local cytokines.

The expression "mechanical hyperalgesia" is still present in the manuscript (abstract, description of the results, discussion) when the authors refer to von Frey's results. The authors have to change this expression since the up and down method with von Frey allows to assess tactile allodynia (or mechanical pain hypersensitivity as suggested in the first revision).

Reviewer #2: This is the R1 version of the MS of Miho Asahara and colleagues.

Major points:

-In the answer: “Furthermore, according to the reference mentioned below (doi: 10.1186/s12990-015-0010-9), MPO-rich Ly6Chigh monocytes promote inflammation and tissue destruction via the release of proteases and inflammatory cytokines, such as TNF-α, in various inflammatory settings.” The cited reference does not show what was mentioned.

-Previous question was: The discrepancy between the present finding and the previous finding (DOI:10.1073/pnas.1501372112) should be better explored with experiments. These papers are suggesting different cellular populations are involved in incisional pain. What are Ly6Chigh inflammatory monocytes doing in incisional pain and how LTB4 can affect their function? What is the explanation for the difference at day 1? At this time point there is no significant analgesia by BLT1 deficiency, but there is reduction of Ly6Chigh inflammatory monocytes and IL-1b and TNFa production. However, Ly6Chigh inflammatory monocytes are high at day 3 and we do not know the cytokine profile.

Answer was: Previous reports from other group (DOI: 10.1073/pnas.1501372112) have shown that monocytes (nonneutrophil myeloid cells) are more involved than neutrophils in postoperative pain. In our study, we showed that inhibition of monocytes by suppressing BLT1 signaling also contributes to pain control. BLT1-deficiency resulted in the suppression of monocyte infiltration on days 1 and day 3, and may also be involved in the suppression of cytokines on day 1.

Regarding the reason why the pain response in BLT1KO was not suppressed on day 1,

we speculate that inflammatory reaction was relatively strong on day 1; thus, the stronger monocyte suppression may be necessary for pain inhibition.

The data of cytokine production at day 3 was added to the new Fig 4.

Comment: Authors did not answer the question, but rather tried to blur what was questioned. The finding (DOI:10.1073/pnas.1501372112) disproved the contribution of inflammatory (CCR2(+)Ly6C(hi)) monocytes in post-incisional pain. The current Authors are working to demonstrate that Ly6Chigh inflammatory monocytes are contributing to post-incisional pain since BLT1 deficiency reduced their presence in the tissue and pain. In addition to this dismissed explanation of the Authors, Ly6Chigh inflammatory monocytes reduced when there was no reduction of pain, and again, Authors do not give a reasonable explanation, but rather speculate without data.

-As mentioned by Reviewer #1, this MS needs more pain measurements. Significant reduction of allodynia was observed at days 2, 3 and 4. However, LTB4 production was significant only at 3h and day 1. LTB4 activates TRPV1, thus, there is need to evaluate thermal hyperalgesia.

-Another point that might be happening is that the Authors are not looking at the right cell type. Authors must assess CD11b(+)Ly6G(-) myeloid cells as previously demonstrated (DOI:10.1073/pnas.1501372112). This might be the reason why the results are not aligning.

-Figure 4. Two cytokines known to induce pain (TNFa and IL-1b) have their production reduced by BLT1 deficiency at day 1 only, again, when there is no change in pain. I understand that some mediators cause nociceptor sensitization, however, this can be seen within some hours and not skipping one entire day. The time points of measurement after incision might also be influencing the results.

-Authors also mention that they are presenting the individual results to allow verifying the variability but did not answer the question on whether results are normal and homogeneous.

Reviewer #3: The authors have responded appropriately to this reviewer’s comments.

However, the basal 50% paw withdrawal threshold shown in Figure 5 as an additional experiment to determine the effect of intraplantar injection of LTB4 (about 1.5 g) is quite different from that in Figure 1 (about 3.6-3.7 g). The authors should comment to this difference. Is the experimenter performing von Frey filament test who was blinded to the treatment and genotypes of mice one and the same person?

7. PLOS authors have the option to publish the peer review history of their article (what does this mean?). If published, this will include your full peer review and any attached files.

Reviewer #1: **Yes: **Cyril Rivat

Reviewer #2: No

Reviewer #3: No

---

## [Author Response · Author response to Decision Letter 1]

16 Sep 2022

We appreciate all reviewers for reviewing our manuscript and giving useful comments to improve our manuscript again. 

Additional Editor Comment:

Because of the unavailability of the original editor, I am handling your submission.

Your revised manuscript has been evaluated by three reviewers. Though two of them are more positive, Reviewer #2 raised serious concerns. In particular, as the reviewer pointed out, the results reported in 10.1073/pnas.1501372112 do not support the notion that CCR2+Ly6Chi cells contribute to pain. Accordingly, the mechanistic role of Ly6Chi monocytes in pain in your model needs to be further addressed, and your findings need to be better correlated with the literature.

Reviewers' comments:

Reviewer's Responses to Questions

Review Comments to the Author

Reviewer #1: The authors did an excellent work in addressing my comments and those raised by the different reviewers. The changes made by the authors greatly improve the manuscript. The remaining comments should be considered by the authors

Regarding the LTB4 plantar injection, a suggestion would be to evaluate cytokine concentration in plantar tissue to support the peripheral action of LTB4 on local cytokines.

Measurements of cytokine concentration in plantar tissue following LTB4 injection were performed previously. However, IL-1β, TNF- α and IL-6 elevation were small and not significantly different from vehicle injection group. (L298-300). The increase in LTB4 may not be directly related to local cytokine production. 

The expression "mechanical hyperalgesia" is still present in the manuscript (abstract, description of the results, discussion) when the authors refer to von Frey's results. The authors have to change this expression since the up and down method with von Frey allows to assess tactile allodynia (or mechanical pain hypersensitivity as suggested in the first revision).

We checked again and replaced the term “Mechanical hyperalgesia” by “Mechanical pain hypersensitivity”.

Reviewer #2: This is the R1 version of the MS of Miho Asahara and colleagues.

Major points:

-In the answer: “Furthermore, according to the reference mentioned below (doi: 10.1186/s12990-015-0010-9), MPO-rich Ly6Chigh monocytes promote inflammation and tissue destruction via the release of proteases and inflammatory cytokines, such as TNF-α, in various inflammatory settings.” The cited reference does not show what was mentioned.

I apologize for citing the wrong reference. The correct reference is doi: 10.1084/jem.20070885.

-Previous question was: The discrepancy between the present finding and the previous finding (DOI:10.1073/pnas.1501372112) should be better explored with experiments. These papers are suggesting different cellular populations are involved in incisional pain. What are Ly6Chigh inflammatory monocytes doing in incisional pain and how LTB4 can affect their function? What is the explanation for the difference at day 1? At this time point there is no significant analgesia by BLT1 deficiency, but there is reduction of Ly6Chigh inflammatory monocytes and IL-1b and TNFa production. However, Ly6Chigh inflammatory monocytes are high at day 3 and we do not know the cytokine profile.

Answer was: Previous reports from other group (DOI: 10.1073/pnas.1501372112) have shown that monocytes (non-neutrophil myeloid cells) are more involved than neutrophils in postoperative pain. In our study, we showed that inhibition of monocytes by suppressing BLT1 signaling also contributes to pain control. BLT1-deficiency resulted in the suppression of monocyte infiltration on days 1 and day 3, and may also be involved in the suppression of cytokines on day 1.

Regarding the reason why the pain response in BLT1KO was not suppressed on day 1,

we speculate that inflammatory reaction was relatively strong on day 1; thus, the stronger monocyte suppression may be necessary for pain inhibition.

The data of cytokine production at day 3 was added to the new Fig 4.

Comment: Authors did not answer the question, but rather tried to blur what was questioned. The finding (DOI:10.1073/pnas.1501372112) disproved the contribution of inflammatory (CCR2(+)Ly6C(hi)) monocytes in post-incisional pain. The current Authors are working to demonstrate that Ly6Chigh inflammatory monocytes are contributing to post-incisional pain since BLT1 deficiency reduced their presence in the tissue and pain. In addition to this dismissed explanation of the Authors, Ly6Chigh inflammatory monocytes reduced when there was no reduction of pain, and again, Authors do not give a reasonable explanation, but rather speculate without data.

According to the FACS data of immune cells of planter tissue (DOI:10.1073/pnas.1501372112), Ly6Chigh cell subset peaked on day1 and declined on day3. On the other hand, Ly6Chigh cell subset increased on day1 and peaked on day3 in this study, suggesting that intensity of inflammation may be stronger than their model and the effect of Ly6Chigh cell subset appeared more potently. We have described this in the text (L348-353).

-As mentioned by Reviewer #1, this MS needs more pain measurements. Significant reduction of allodynia was observed at days 2, 3 and 4. However, LTB4 production was significant only at 3h and day 1. LTB4 activates TRPV1, thus, there is need to evaluate thermal hyperalgesia.

-Another point that might be happening is that the Authors are not looking at the right cell type. Authors must assess CD11b(+)Ly6G(-) myeloid cells as previously demonstrated (DOI:10.1073/pnas.1501372112). This might be the reason why the results are not aligning.

Previous literature reported that CD11b+Ly6G- myeloid cells are required for the mechanical hypersensitivity not for the thermal hypersensitivity that follows incisional wound-induced inflammation (DOI:10.1073/pnas.1501372112). In this study, we also focused on the role of inflammatory cells and LTB4-BLT1 signaling in the mechanical pain hypersensitivity following incision. 

CD11b+Ly6G- myeloid cells are assessed by analyzing the cell population after eliminating CD11b+Ly6G+ cells. Cell suspensions were first gated on forward-scatter (FSC-A) and side-scatter (SSC-A) to exclude debris. Neutrophils were then selected as double-positive for CD11b and Ly6G, quantified, and eliminated from further analysis. Non-neutrophil myeloid cells (CD11b+Ly6G-) were then gated based on Ly6C expression as Ly6Chigh monocytes or Ly6Clow monocytes. Representative FACS plots of at 3day time point after incision from WT mice are shown in the below figures (Please see another response to reviewers document). Thus, CD11b+Ly6G+ myeloid cells (Neutrophil) and CD11b+Ly6G-Ly6C+ non-neutrophil myeloid cells (monocyte) were analyzed. In this study, FACS analysis of CD11b+Ly6G+ myeloid cells were shown in Figure 3 A and C. CD11b+Ly6G+ myeloid cells peaked on day1, and no significant difference was observed between BLT1WT and BLT1KO. On the other hand, CD11b+Ly6Chigh subset were significantly reduced in BLT1KO mice than BLT1WT mice on day1 and day3 (Figure 3 B and D). 

Ghasemlou et.al. showed that CD11b+Ly6G+ neutrophils do not contribute to the development of incisional pain by CD11b+Ly6G+ cell depletion study. Furthermore, depletion of CD11b+Ly6G- cells were confirmed by CD11b-TK /GCV mice and these treatments showed significant increase of mechanical threshold. (DOI:10.1073/pnas.1501372112). Since most Ly6G- cells (=Ly6C+) disappeared in this cell-depletion treatment, it was difficult to distinguish which subset group (Ly6Clow or Ly6Chigh) was responsible for the incisional pain. They tried to see the effect of depletion of Ly6Chigh and CCR2KO mice were used. In CCR2KO, reduction in all three subsets of Ly6C were shown; Ly6Clow subset displayed a much smaller reduction at 24h than 3d. However, there was no significant difference in incisional pain responses between CCR2WT and CCR2KO.

Their results of the recruitment of CD11b+Ly6G- cells contribute to incisional pain is aligning our results, but whether or not which subset of CD11b+Ly6C+ cells were responsible in incisional pain was remains to be elucidated. In this study, recruitment of the CD11b+Ly6Chigh subset were reduced in BLT1KO and could be involved in the attenuation of pain response. 

-Figure 4. Two cytokines known to induce pain (TNFa and IL-1b) have their production reduced by BLT1 deficiency at day 1 only, again, when there is no change in pain. I understand that some mediators cause nociceptor sensitization, however, this can be seen within some hours and not skipping one entire day. The time points of measurement after incision might also be influencing the results.

We consider that the reduction of the two cytokines reflects the suppression of monocyte infiltration by BLT1KO. However, whether a transient early increase of two cytokines is involved in the pain response cannot be resolved in this study. We speculate that the inflammatory reaction was relatively strong on day1, and, thus, stronger monocyte suppression may be necessary for pain inhibition during this period. Furthermore, in this study, neutrophil recruitment did not differ between KO and WT and peaked on day1, suggesting that other inflammatory mediators and chemokines released from neutrophils may sensitize peripheral nociceptors on day1. Description of peak time of neutrophil infiltration was corrected (L252).

-Authors also mention that they are presenting the individual results to allow verifying the variability but did not answer the question on whether results are normal and homogeneous.

Analysis of Normality were performed by Shapiro-Wilk test. Homogeneity of variance was confirmed by the Leven test using R statistical software version 4.2.1(R project for statistical computing). Description was added in the Method section. (L298-300)

Reviewer #3: The authors have responded appropriately to this reviewer’s comments.

However, the basal 50% paw withdrawal threshold shown in Figure 5 as an additional experiment to determine the effect of intraplantar injection of LTB4 (about 1.5 g) is quite different from that in Figure 1 (about 3.6-3.7 g). The authors should comment to this difference. Is the experimenter performing von Frey filament test who was blinded to the treatment and genotypes of mice one and the same person?

The mice used for behavior test in Figure 5 were 8 weeks old, and younger than the mice used in Figure 1(9-14 weeks old). It is possible that the age of the mice used affected the baseline threshold.

The experiments of assessment of pain behavior were performed in a blinded manner to the treatment and genotypes of mice.

---

## [Decision Letter · Decision Letter 2]

29 Sep 2022

Role of leukotriene B4 (LTB4)-LTB4 receptor 1 signaling in post-incisional nociceptive sensitization and local inflammation in mice

PONE-D-22-08079R2

Dear Dr. Ito,

We’re pleased to inform you that your manuscript has been judged scientifically suitable for publication and will be formally accepted for publication once it meets all outstanding technical requirements.

Kind regards,

Ichiro Manabe

Academic Editor

PLOS ONE

Additional Editor Comments (optional):

Reviewers' comments:

Reviewer's Responses to Questions

**Comments to the Author**

1. If the authors have adequately addressed your comments raised in a previous round of review and you feel that this manuscript is now acceptable for publication, you may indicate that here to bypass the “Comments to the Author” section, enter your conflict of interest statement in the “Confidential to Editor” section, and submit your "Accept" recommendation.

Reviewer #1: (No Response)

Reviewer #3: All comments have been addressed

2. Is the manuscript technically sound, and do the data support the conclusions?

Reviewer #1: Yes

Reviewer #3: Yes

3. Has the statistical analysis been performed appropriately and rigorously? 

Reviewer #1: Yes

Reviewer #3: Yes

4. Have the authors made all data underlying the findings in their manuscript fully available?

Reviewer #1: Yes

Reviewer #3: Yes

5. Is the manuscript presented in an intelligible fashion and written in standard English?

Reviewer #1: Yes

Reviewer #3: Yes

6. Review Comments to the Author

Reviewer #1: The authors addressed the remaining issues except that in the abstract lines 29 and 33, the term "Mechanical hyperalgesia" should still be changed.

Reviewer #3: (No Response)

7. PLOS authors have the option to publish the peer review history of their article (what does this mean?). If published, this will include your full peer review and any attached files.

Reviewer #1: No

Reviewer #3: No

---

## [Editor Report · Acceptance letter]

10 Oct 2022

PONE-D-22-08079R2 

Role of leukotriene B4 (LTB4)-LTB4 receptor 1 signaling in post-incisional nociceptive sensitization and local inflammation in mice 

Dear Dr. Ito:

I'm pleased to inform you that your manuscript has been deemed suitable for publication in PLOS ONE. Congratulations! Your manuscript is now with our production department. 

Kind regards, 

on behalf of

Dr. Ichiro Manabe 

Academic Editor

PLOS ONE